Shapes of leaves with parallel venation. Modelling of the Epipactis sp. (Orchidaceae) leaves with the help of a system of coupled elastic beams

Jakubska-Busse Anna anna.jakubska-busse@uwr.edu.pl 1
Janowicz Maciej 2
Ochnio Luiza 2
Jackowska-Zduniak Beata 2
1 Department of Botany, Institute of Environmental Biology, University of Wrocław , Wrocław , Poland
3 Faculty of Applied Informatics and Mathematics, Warsaw University of Life Sciences–SGGW , Warsaw , Poland
Nie Qing
Electronic publication date: 2016 Jun 28
Publication date: 2016
Volume: 4
Electronic Location ID: e2165
Received 2016 Mar 17; Accepted 2016 Jun 2
Copyright: ©2016 Jakubska-Busse et al.
Copyright year: 2016
Copyright holder: Jakubska-Busse et al.
License: This is an open access article distributed under the terms of the Creative Commons Attribution License, which permits unrestricted use, distribution, reproduction and adaptation in any medium and for any purpose provided that it is properly attributed. For attribution, the original author(s), title, publication source (PeerJ) and either DOI or URL of the article must be cited.
License URL: https://creativecommons.org/licenses/by/4.0/

Keywords: Plant architecture, Coupled elastic beams, Mathematical biology, Plant modelling, Elasticity, Leaves, Orchids, Epipactis, Undulation, Modelling plant species

Funding: The authors received no funding for this work.

==============================
Static properties of leaves with parallel venation, with particular emphasis on the genus EpipactisZinn, 1757 (Orchidaceae, Neottieae) have been modelled with coupled quasi-parallel elastic “beams.” The non-linear theory of strongly bended beams have been employed. The resulting boundary-value problem has been solved numerically with the help of the finite-difference method. Possible dislocations resulting in additional Dirac-delta like forces have been take into account. Morphological similarity of the model and real leaves has been obtained. In particular, the concentrated forces have been shown to cause undulation in the leaves.

Introduction

The modelling of plant organs remains an open problem due to the complexity of plant architecture, regardless of the particular organ or any particular taxonomical group of plants being considered. In this study, we are concerned with the leaves of Orchidaceae and specifically, the leaves of plants belonging to the genus Epipactis.

Orchidaceae are a varied group that includes both terrestrial plants and epiphytes with monopodial and sympodial growth, which differ from each other by leaf construction in both shape and size. Most orchid leaves are typical of the monocots with many parallel veins and the cross connections between the longitudinal veins being inconspicuous (Dressler, 1981; Jakubska-Busse & Gola, 2014). An orchid leaf usually consists of only a blade, though some species produce also the petiole which may be narrow or which may widen and embrace a stem, forming a leaf sheath. The shape of the blade tends to be modified alongside its size as a result of growth processes (Jakubska-Busse & Gola, 2014). Orchid taxa may produce leaves of different shapes, e.g., plicate and cordate, plicate with a sheathing base, convolute, non-articulate, conduplicate and deeply lobed or, rarely, twisted (Dressler, 1981).

An interesting and relatively well-studied issue is leaf undulation, which is often observed in monocots including orchids. It has been found that the waves in the leaf blades in monocots usually appear perpendicular to the leaf length, which demonstrates that as the leaf surface grows, it changes correspondingly lengthwise to the leaf blade. It is worth noting that although undulation normally occurs perpendicularly to the leaf length, at the beginning stage it occurs alongside its length. Displacements in the wrinkled leaf also occur across the leaf blade. Whichever way they appear, they demonstrate that the pace of their growth is irregular since wrinkles in a leaf can be of different length. Hejnowicz (1991) found that spatial and temporal fluctuations in the pH of the epidermal cell walls aided the undulation.

Epipactis as the main object of our study is a mainly Eurasian genus with a south-central distribution (Rasmussen, 1995; Delforge, 2006). These are all rhizomatous (clonal), summer-green plants with habitats ranging from bogs to dry forest (Delforge, 2006). There are some controversies around the taxonomy as one of the Epipactis muelleri Godf. species is identified based on the undulation of its leaf blade, alongside gynostemium features. This property is quoted in nearly every artificial key for determining this group of orchids, e.g., Sundermann (1975), Procházka & Velísek (1983), Potůček & Čačko (1996) and Delforge (2006). However, it has been shown by Jakubska-Busse & Gola (2014) that the leaf undulation in Helleborines does not have any diagnostic value as a non-programmed intrinsic feature and should not be applied to taxa identification. In this paper, we attempt to show which characteristics in the mathematical model of leaf are responsible for the undulation.

Material and Methods

In order to build the model, we have preliminarily checked the fulfillment of Hooke’s law on a sample of 72 fresh Epipactis sp. (E. helleborine, E. muelleri, E. albensis and E. palustris) leaves. Those measurements have been used here to justify the approach based on theory of elasticity as modeling tool. (We plan to perform much more extensive measurements in some future.) A code written in the Python language was used to solve the model described below and this employed the finite-difference method to solve boundary-value problem for a system of ordinary differential equations. We have also used published data on the morphology and anatomy of the Epipactis sp. leaves (Jakubska-Busse & Gola, 2010; Jakubska-Busse & Gola, 2014; Jakubska-Busse et al., 2012). The presented studies were done with the consent of the Regional Director for Environmental Protection, No. WPN.6400.27.2015.IW.1.

Results

The model

The modelling of soft materials, in particular organic ones, is a notoriously difficult problem. Indeed, any functional organic tissue usually contains millions of elements, namely the cells and furthermore, such tissue is an arena of very complex chemical reactions combined with diffusion. Needless to say, the above statements also hold in the case of plant leaves. There are at least two possible theoretical approaches to the modelling of leaves. Firstly, one can think about simply simulating the shapes of the leaves and secondly, one can attempt to start (almost) from the first principles and include the cellular structure of the leaf tissue. With regard to plant morphology in general, including that of leaves, tremendous progress has been recently achieved within the framework of L-systems. Furthermore, the almost realistic pictures in two dimensions can be obtained by application iterated functions in a similar way to the construction of the Mandelbrot set (Bird & Hoyle, 1994). Bird and Hoyle’s simple approach has, however, nothing to do with any physical, chemical and biological properties of living organs. A more sophisticated approach has been developed using iterative functions by Runions et al. (2005) for the modelling of leaf venation. A powerful method to model biological pattern formation is the numerical solution of various reaction–diffusion equations (Koch & Meinhardt, 1994). A method to study general biological systems which exhibit various line-like and net-line structures using coupled reaction–diffusion equations was developed in the seventies (Meinhardt, 1976). Markus, Böhm & Schmick (1999) made an interesting attempt to model reaction–diffusion equations using cellular automata and then applying them to vessel morphogenesis including leaf venation. More recently, the shapes of leaves have been modelled by this method by Franks & Britton (2000). Another interesting contribution based on hydrodynamics rather than reactions and diffusion has involved the numerical solutions to the Navier–Stokes equations (Wang, Wan & Baranoski, 2004). Another starting point in leaf simulation has been worked out in Liang & Mahadevan (1999). The shapes of long leaves have been obtained using the standard theory of elasticity. Saddle-like midsurface and rippled edges of leaves have been interpreted as being caused by elastic relaxation through the bending that follows differential growth. The first-principles line of dealing with the problem of leaf modelling is still in statu nascendi; there is, however, promising work on the simulation of plant tissues (Ghysels et al., 2009; Van Liedekerke et al., 2011) based on micromechanical models of single cells (Van Liedekerke et al., 2010). Our approach lies in between a purely phenomenological approach and one based on the cell micromechanics. We realise that in the case of long leaves with parallel venation the anisotropy and inhomogeneity are essential, and models which assume isotropy and/or homogeneity must fail. This contrasts our model with that of Liang & Mahadevan (1999). However, we agree that elasticity theory is an appropriate framework for the description of leaf morphology. Indeed, regardless of what happens in the cells and what interaction between cells may be detected, the result must always involve changes in purely mechanical quantities which characterize a solid material. A continuum-mechanical model makes it possible to relate the elastic properties of the model with those of real leaves. We note here that the experimental research on the mechanical properties of leaves appears to have only just begun, and we are aware of only a single comprehensive study of such properties (Van Liedekerke et al., 2010). We have performed a couple of experiments (to be reported elsewhere), in which we have measured the dependence of the length of Epipactis sp. leaves on the applied elongating force. We found that the assumption of the validity of Hooke’s Law (i.e., the linear dependence) is satisfactorily fulfilled; its breakdown happens if the applied forces are close to those which destroy the leaves. The results of our experiments encouraged us to consider the model of a leaf based on a simple version of the theory of elasticity. Our main assumptions in the building of the model are the following: (i) the shape of a leaf is determined by the distribution of its veins, (ii) each vein, together with its surrounding tissue, can be represented by an elastic beam, the shape of which is given by the non-linear theory of strongly deformed beams with circular cross-sections (Landau & Lifshitz, 1993), (iii) the veins are elastically coupled with their nearest neighbours due to the presence of the tissue between the veins, and (iv) only the main (“first-order”) veins are taken into account explicitly whilst the presence of secondary veins may lead to additional concentrated forces acting upon the principal veins. According to our hypothesis, the characteristic undulation near the edges of a leaf is a result of dislocations both in the regions between the veins and in both primary and secondary veins.

In our opinion, the above model has the following justification from the point of view of the leaves micromorphology. In the paper (Jakubska-Busse & Gola, 2014) it has been demonstrated that “due to changes in the number of cell rows (by cell division) and cell sizes/volumes (by cell growth), repeated along the edge of the leaf, sectors with local expansion of the surface affected the entire leaf blade structure” (Jakubska-Busse & Gola, 2014). The resulting “local expansion” can and actually should be physically interpreted as “a defect” in the structure of the beams in the corresponding model. This is because the growth in the number of cells leads to the clear perturbation of the cell sequences—that is, additional, quasi-parallel cell sequences appear rather abruptly. Similar effects appear on the veins themselves, please see additional discussion in ‘Discussion.’

Let dlm be the infinitesimal element of the arc along the mth beam, and let tm be the unit vector tangent to that beam. The shape of the beam is given by the function rlm such that drdlm=tm.

Let Fm denote the force which characterizes the internal stress in the beam. If Km denotes the external forces (per unit length) which act upon the beam, F has to satisfy (Landau & Lifshitz, 1993): (1) dFmdlm=−Km.

On the other hand, the rate of change of the moment Mm associated with the force Fm along the beam can be written as: (2) dMmdlm=Fm×tm.

For the beam of circular cross section the moment Mm can be written as: (3) Mm=EmImtm×dtmdlm,

where Em is the Young modulus of the mth beam, and Im(z) is its area moment of inertia.

It follows that the tangent vector changes along the beam according to: (4) EmImtm×d2tmdlm2=Fm×tm.

Let us obtain the equations for the components of the vector tm in the spherical coordinates, that is, let tm=cosθmcosϕm,cosθmsinϕm,sinθm.

Computation of the cross products leads to (please see the Maxima file contained in Supplemental Information 1 for derivation): (5) EmImcosθmd2ϕmdlm2−2sinθmdθmdlmdϕmdlm=Fx,m sinϕm−Fy,m cosϕm

and (6) EmImd2θmdlm2+12sin2θmdϕmdlm2=Fx,m sinθmcosϕm+Fy,m sinθmsinϕm−Fz,m cosθm.

The above equations have to be supplemented with the the relation: (7) drdlm=tm,

and the expression for the components of the force Fm: (8) dFi,mdlm=−Ki,m,

where i = x, y, z. We can now define a new dimensionless parameter s such that dlm = Lms with Lm being the length of the mth beam. In the following we also use rescaled (dimensionless) forces Gi,m such that: Fi,m=EmImLm2Gi,m.

It is also convenient to work with dimensionless components of the coordinate vector: (9) ξm=xm∕Lm,ηm=ym∕Lm,ζm=zm∕Lm.

Then the shape of the mth beam together with internal forces are completely specified if we solve the system: (10) cosθmd2ϕmds2−2sinθmdθmdsdϕmds=Gx,m sinϕm−Gy,m cosϕm,

(11) d2θmds2+12sin2θmdϕmds2=Gx,m sinθmcosϕm+Gy,m sinθmsinϕm−Gz,m cosθm.

From the definition of the tangent vector tm we also have: (12) dξmds= cosθmcosϕm,

(13) dηmds= cosθmsinϕm,

(14) dζmds= sinθm.

We have assumed the following simple form of the force K ¯m=Lm3∕EmImKm: (15) K ¯i,m=−κi,m2ρi,m−ρi,m+1−ρi,m−1−qi,mδi,z,

where i = x, y, z and qi,m = qmδiz is the scaled, dimensionless weight of the mth beam per unit length. Therefore, the equations for the dimensionless forces Gi,m take the form: (16) dGi,mds=κi,m2ρi,m−ρi,m+1−ρm−1−qmδi,z.

In the following we have assumed that there is no external force with non-vanishing x coordinate acting on any beam except, perhaps, of that acting at the tip of the leaf. Thus, the above system of differential equations simplifies a bit since Gx,m is independent of s. The resulting system of differential equations has the order 9M and its solution is available only numerically. The boundary conditions has been chosen as follows. We have used ηm(0) = ζm(0) = 0, ξm(1) = 1, ηm(1) = ζm(1) = 0, Gy,m(1) = Gz,m(1) = 0. We have also prescribed ϕm(0) = ϕm0, θm(0) = θm0 and experimented with various ϕm0 and θm0. The above boundary conditions correspond to the left (s = 0) end of each beam lies on the ξ-axis. About its right end (s = 1) one can say that the sum of all forces acting on the beams there vanishes. As the tip can be considered point-like, the beams meet at one point. We can assume (performing rotation of the coordinate system if necessary) that the tip is located at (1, 0, 0), hence the boundary conditions for ξm(1), ηm(1), ζm(1). The above boundary condition effectively imply that we have to do with the buckling of beams in our model due to the concentrated forces very close to the tip of the leaf. That buckling clearly influences, to some extent, the shape of the blade as well as the wrinkling. Ideally, we should have prescribed a (possibly much more complicated) correct model of the inter-beam forces. The convergence of the beams near the tip would then be a consequence of such a good model. Unfortunately, we have encountered serious difficulties in finding such a convincing description. So far, we have decided to introduce the tip simply by the above boundary conditions. It seems to us, however, that the presence of strong concentrated forces near the end of the blade is not unlikely, and that the resulting buckling does influence the shape of the leaf. Let us notice that the ξ coordinates of the left ends of the beams are unspecified and they actually result from the simulation. They are always close to, but never exactly equal to 0. This results in appearance of a small “petiole.”

In addition to forces specified above, we have assumed the presence of concentrated, Dirac-delta like forces in order to model dislocations (this is consistent with the elementary theory of dislocations as described in Landau & Lifshitz (1993)). That is, we have added the forces of the form fz,ms= ∑kgm,kδs−sk

to the right-hand sides of Eq. (16). Only the z-component of (16) has been modified this way.

Apart from gravity, the total external force which acts on the beams must naturally be equal to zero.

Numerical Results

The above boundary-value problem has been solved using its representation in terms of four second-order equations and finite differences. The discretization has been performed with N = 101 points and si − si−1 = :h = 1∕(N − 1). It has been assumed as a kind of “zeroth-order approximation” that all qm are the same, qm = q = − 1.0.

The coefficients g(m, k) have been assumed on m as follows: gm,k=g0k1− cosπ∕Mm−M+1∕2

so that the (scaled) internal concentrated forces are larger near the edges and smaller in the central part of the leaf. Since these forces are scaled, that difference can be attributed either to forces themselves, or to smaller Young modulus and/or area inertial moments of the lateral beams.

Figure 1 Examples of the shapes of leaves for vanishing g0, i.e., without the presence of point-like internal forces.

The following values of parameters have been used: q0 = − 1.0, θm(0) = 0, m = 1, 2, …, M, κy = 10.0, κz = 10.0, g0 = 0.0. (A) Shapes of the beams which form the leaf structure. (B) Projection of the leaf of (A) on the ξ − η plane.

Figure 2 The same as in Fig. 1 but for different coupling between the beams: κy = 10.0, κz = 5000.0.

There are no point-like internal forces, i.e., g0 = 0.0.

Figure 3 Examples of the shapes of leaves in the presence of Dirac-delta like forces caused by defects in the leaf structure (i.e., non-vanishing g0).

The following values of parameters have been used: q0 = − 1.0, θm(0) = 0, m = 1, 2, …, M, κy = 10.0, κz = 10.0, s1 = 0.1, s2 = 0.3, s3 = 0.5, s4 = 0.6, s5 = 0.7, s6 = 0.9, g0(1) = − g0(2) = 50, g0(3) = − g0(4) = − 250, g0(5) = − g0(6) = 150.

Figure 4 Examples of the shapes of leaves in the presence of Dirac-delta like forces caused by defects in the leaf structure (i.e., non-vanishing g0).

The following values of parameters have been used: q0 = − 1.0, θm(0) = 0, m = 1, 2, …, M, κy = 10.0, κz = 10.0, s2i−1 = i∕10, s2i = (i + 1∕2)∕10, i = 1, 2, …, 6; g0(1) = g0(3) = − g0(2) = − g0(4) = 500, g0(5) = g0(7) = g0(9) = g0(11) = − g0(6) = − g0(8) = − g0(10) = − g0(12) = 400. The parameters correspond to strong defects or a small Young’s modulus close to the edges, and it results in stronger undulations in those regions.

Figure 5 Natural forms of the Epipactis leaves.

Photo by Anna Jakubska-Busse.

We have worked with M = 32 beams and used the finite differences to discretize the differential operators. Since the resulting matrices in our boundary-value problem have been quite large, we have not used any solver, but employed the method of false transients instead. The results of these simulations are illustrated in Figs. 1–4 for four sets of parameters. Figures 1A, 2A, 3A and 4A display the shapes of all beams. Figures 1B, 2B, 3B and 4B show the projections of the blades of leaves onto the ξ − η plane.

Figure 1 displays the results for a leaf under the absence of any defects (hence, no point-like forces appear) and small coupling between the beams which form the leaf blade.

In Fig. 2 we have shown a shape which has resulted from much stronger coupling between the beams, but still under the absence of any point-like forces. The difference between the shapes in Figs. 1 and 2 is evident, but there is no undulation.

We have performed preliminary measurements of the amplitudes of wrinkles for several leaves. It has been found that the amplitudes of the leaf wrinkles have normally been of the order of 1–2% (it’s depended of course on whether we measured in the apical, central or basal part of the leaf) but not exceeded 10% of its length The parameters of the model are chosen in such a way as to fit that maximum value of the relative (i.e., related to the length of the leaf) amplitudes. In Fig. 3, however, the number of defects as well as the strengths of point-like forces are too small to cause any considerable wrinkles. Only in Fig. 4 are they strong enough to produce a clear picture of undulation. As we observed before, large values of the parameters g0(i) can be attributed to both strong defects and to small values of the Young modulus of the beams. In particular, smaller values of the Young modulus close to edges results in the stronger undulations in those regions. Please see Fig. 5 for comparison.

Discussion

In this paper the model of a plant leaf with quasi-parallel venation (as exemplified by the Epipactis sp. leaf) which consists of coupled elastic beams has been developed. Only the nearest-neighbour interactions between the beams have been employed. It was found that the model reproduces quite well the overall shape of the leaf. In particular, it reflects to some extent the presence of undulation at the leaf edges. The appearance of undulation is attributed to the following factors: (a) a small Young’s modulus and/or moment of inertia of veins which are close to the edges; (b) dislocations in the structure of the principal veins themselves and/or connecting tissue and secondary veins which result in the large but well-localized additional forces which act on the veins; those dislocation appear during the growth of the leaf.

The presence of the defects in the sequences of cells in the space between veins has been discussed in (Jakubska-Busse & Gola, 2014). As for the presence of clear change of such sequences close to the veins themselves, let us just invoke Fig. 6 where it seems to be transparent.

Figure 6 SEM (scanning electron microscope) photography of the upper (adaxial) leaf surface of Epipactis sp.

Sequences of cells along a vein. Change of the pattern from a single row of cells (indicated by red arrow) to three rows (indicated by green arrow) is well visible.

Some of the shortcomings of the analysis in this paper are the following.

Firstly, even though the theory of dislocations in elastic bodies predicts the presence of Dirac-delta-like forces, it is entirely unclear whether they could act in precisely the way we have described above. Detailed microscopic observations together with thorough and complex micromechanical models could bring the answer. Quite obviously, we have applied the Dirac-delta-like forces in an ad hoc manner to obtain numerically results close to those known in real leaves.

Secondly, we have assumed a lot (and possibly too much) about our beams to simplify the model. Needless to say, the cell sequences in the leaves do not have circular cross sections. It may very well happen that this fact, together with strong forces acting near the apex of a leaf, is a source of instability which is and additional factor in formation of the undulation. Nonetheless, we believe that our arguments based on the findings of Jakubska-Busse & Gola (2014) and Fig. 6 provide first necessary steps in the proper direction.

In some remarkable recent works (Marder, 2003; Audoly & Boudaoud, 2004; Sharon, Roman & Swinney, 2007), the variety of shapes of leaves in general, and their wrinkling in particular, has been interpreted in a very elegant way, namely, in terms of Riemannian geometry of surfaces. What is more, those authors have described the effect of applying auxin at leaf edges to produce ripples and emphasize more explicitly growth in the leaf. It is conjectured that in the developmental processes of naturally growing tissues, the process of growth provides a mechanism for the spontaneous formation of non-Euclidean metrics and consequently leads to complicated morphogenesis of thin films (including surfaces of leaves) exhibiting waves, ruffles and non-zero residual stress (Lewicka, 2011). What we have attempted here is to provide a more physical, granular model which is from the very beginning adapted to the strong anisotropy of Epipactis leaves and concerned with local forces and torques. We believe that, to some extent, our model is complementary to that developed in the papers quoted above, being, perhaps, somewhat closer to the microscopic reality of the Epipactis leaves. An interesting question which arises in this connection is whether we could appropriately interpret within our model the results of the experimentation with auxins. Let us first stress that the leaves used in the those experiments (daffodils (Narcissus)) are quite different from those of Epipactis. Nevertheless, our model seems to be sufficiently abstract to justify its application to other leaves with both similarly shaped leaves and similar arrangements of main veins. The outcomes of experiments with auxins could be explained within our model as resulting from the (local) change of physical characteristics of the beams, especially their area moments of inertia. We also feel that the departure from the assumption of the circularity of cross sections of the beams might be necessary.

Further research is necessary in at least two directions. Firstly, the experimental data regarding the mechanical properties of the leaves in general, and the leaves of Orchidaceae in particular, is presently insufficient. Secondly, it is obviously necessary to combine a purely mechanical model with hydrodynamics and the diffusive properties of the dynamics of fluids inside the leaf. We plan to continue our studies in both of these directions.

Supplemental Information

Supplemental Information Data files and Maxima file with derivations of some equations.

Click here for additional data file.

The authors wish to thank Dr. Joanna Ashbourn (University of Oxford) for her suggestions and language correction and Dr. Małgorzata Dudkiewicz (Warsaw University of Life Sciences-SGGW) for helpful discussions. We would also like to thank the anonymous Reviewers for valuable comments on the manuscript.

Additional Information and Declarations

Competing Interests

Author Contributions

Ethics

Data Availability

The authors declare there are no competing interests.

Anna Jakubska-Busse conceived and designed the experiments, analyzed the data, contributed reagents/materials/analysis tools, wrote the paper, prepared figures and/or tables, reviewed drafts of the paper, taxonomic identification of plants (Epipactis).

Maciej Janowicz conceived and designed the experiments, analyzed the data, wrote the paper, prepared figures and/or tables, reviewed drafts of the paper.

Luiza Ochnio and Beata Jackowska-Zduniak conceived and designed the experiments, performed the experiments, analyzed the data, contributed reagents/materials/analysis tools, wrote the paper, reviewed drafts of the paper.

The following information was supplied relating to ethical approvals (i.e., approving body and any reference numbers):

The presented studies were done with the consent of the Regional Director for Environmental Protection, No. WPN.6400.27.2015.IW.1.

The following information was supplied regarding data availability:

The raw data has been supplied as Supplemental Information 1.

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
