# Peer review of "Shapes of leaves with parallel venation. Modelling of the Epipactis sp. (Orchidaceae) leaves with the help of a system of coupled elastic beams"

_PeerJ, doi:10.7717/peerj.2165_

## Round 0.1 · original submission · Major Revisions

Each of the reviewers' comments need to be addressed in detail during your revision.

Reviewer 1 ·

Basic reporting

There are too many errors in mathematical equations (perhaps due to the system when converted to PDF), for instance, the equation between (10) and (12), Equation (15), Line 182, etc. It is so hard to read and check all these formulas (I have to guess), and the authors should be cautious and proof-read the manuscript before submission.

Experimental design

No Comments

Validity of the findings

When compared to real leaves, what is the basis for parameter selection (boundary values)? The paper only shows the morphological similarity between the numerical simulations with real leaves. Is it possible to make any quantitative comparison?

Additional comments

The manuscript studies the properties of leaves with parallel venation by mathematical models with coupled quasi-parallel elastic beams. The model has been numerically solved and the numerical results have been compared with real leaves. It has shown some similarity. There are numerous errors and typos, and also the caption is not well displayed for some figures. Overall the formulas and equations are hard to read and the manuscript is not well written, and the authors should double check and fix technical issues before submission.

Reviewer 2 ·

Basic reporting

The citation Liang and Mahadevan (2009) is good, but later than other works such as references 2,4, and 5 (works of Sharon and colleagues) of the Liang/Mahadevan paper that study undulations at leaf edges. I would suggest citing those papers and comparing the approach and results of those papers with those in this paper. In particular, those authors have described the affect of applying auxin at leaf edges to produce ripples and emphasize more explicitly 'growth' in the leaf. How would the model in this paper apply to an experiment that applies auxin to a leaf edge to induce more growth? Would auxin application change venation patterns or stresses in the veins at a time scale faster than the appearance of undulations? Leaves of plants such as orchids are discussed in the introduction. Is this model more relevant to some plants than others such as the daffodil flower discussed in other literature on undulations in plants?

The English is good. There are a few examples of style that is unusual to me, but maybe less unusual for a PeerJ article. For example,
line 227: "We are, of course, aware of the following shortcomings in the above analysis." I suggest something more like "Some of the shortcomings of the analysis in this paper are the following:" That is, I think it sounds more 'professional' to just state the shortcomings rather explicity defend the authors' judgement in publishing a first model that will need further investigation.

Lines 54-55, "In order to build the model, we have preliminarily checked the fulfillment of Hooke's law on a sample of 72 fresh Epipactis sp. (E. helleborine, E. muelleri, E. albensis and E. palustris) leaves." suggest that the data on the fulfillment of Hooke's law will be reported on in this paper. Later on (line 105), we find out that this will be reported on elsewhere. That's OK, but I suggest making that clear in Lines 54-55.

Experimental design

The research question and the investigative approach are clear.

What takes some work to understand is what the source of the instability that leads to the patterning is.
In the basic assumptions of lines 111-118, the forces that produce an instability come up in assumption iv), but it's not clear to me what the biological source of these forces is. Is it growth in the tissue or some other factor? In Lines 171-174, Dirac-delta-like forces at dislocations are introduced and justified by a reference to Landau and Lifshitz. Again, it's not apparent what the biological/physical source of those forces is--this is a crucial component of the success of the model and the biological meaning of these forces should be discussed.

Validity of the findings

The mathematical formulation and numerical approach are sound. The authors are clear that the model will need further comparison with data but are plausible.

There is a good amount of speculation in the paper, which is well identified as such.

The comment on line 225 of the discussion "a small Young’s modulus and/or moment of inertia of veins which are close to the edges" was not apparent to me when reading through the rest of the paper, except for a small indication in lines 211-214, where I needed to really go back and try to interpret the parameter choices in the simulations. Although parameter values are given for the simulations of Figs 1-4, little help is given to the reader to interpret what each Figure is testing in terms of biological meaning. That is, I suggest for each figure something such as "These parameters correspond to a small Young's modulus close to the edges, and the result is undulations."

Additional comments

This paper can lead to nice work putting together current and future experiments discussed in the paper with simulations.

---

## Round 0.2 · Minor Revisions

Please address the minor comments from the reviewer.

Reviewer 1 ·

Basic reporting

Most of all my previous concerns have been adequately addressed, and I have some minor issues.

1: The sentence "We believe that our arguments based on the findings of Jakubska-Busse and Gola (2014) and 6 provide first necessary steps in the proper direction." has been repeated twice in lines 221 and 225, and I do not quite understand what "6" stands for. Does it mean Figure 6?

2: In the latex, " should be typed as ``. Please correct it through the whole paper.

3: Is it for purpose that all the references miss the numbers?

Experimental design

No further comments

Validity of the findings

No further comments

Additional comments

The authors still need to proof-read the paper and correct some typos.

Reviewer 2 ·

Basic reporting

This has been improved.

Experimental design

Good.

Validity of the findings

The improvements are good.

Additional comments

I am happy with the changes to the manuscript.

---

## Round 0.3 · accepted · Accept

The revision is adequate and your article is now Accepted.